# ExhibitXplorer: Enabling Personalized Content Delivery in Museums Using Contextual Geofencing and Artificial Intelligence

Rosen Ivanov 🄳

Department of Computer Systems and Technologies, Technical University-Gabrovo, 5300 Gabrovo, Bulgaria; rosen@tugab.bg

**Abstract:** In recent years, there has been an increasing demand for personalized experiences in various domains, including the cultural and educational sectors. Museums, as custodians of art, history, and scientific knowledge, are seeking innovative ways to engage their visitors and provide tailored content that enhances their understanding and appreciation of the exhibits. This article presents ExhibitXplorer, a distributed architecture service that leverages geofencing, artificial intelligence, and microservices to enable personalized content delivery in museums. By combining implicit and explicit segmentation of museum visitors and utilizing the GPT API for content generation, ExhibitXplorer offers a dynamic experience to different visitor segments, including researchers, students, casual visitors, and children. The system utilizes push notifications triggered by visitor location changes, allowing seamless delivery of personalized information both indoors and outdoors. Tests were conducted to evaluate the user experience of visitors to an outdoor ethnographic museum. The results showed that 55% of the test participants were satisfied and 45% very satisfied with the way personalized content was delivered.

**Keywords:** personalized content delivery; smart museums; contextual geofencing; artificial intelligence; visitor satisfaction





## 1. Introduction

Museums hold an important place in society as custodians of our collective heritage and as repositories of knowledge. Their importance goes beyond the simple display of artifacts; museums are dynamic educational spaces that play a multifaceted role in the cultural, educational, and therapeutic spheres [1–3].

For centuries, museums have served as living libraries, offering an immersive educational experience that encourages curiosity, critical thinking, and a deeper understanding of the world around us. Museums are invaluable educational institutions that provide an immersive and tangible learning experience. Unlike traditional classrooms, museums provide a multi-sensory environment where students can experience artifacts, artwork, and historical narratives firsthand. They offer a diverse array of educational programs designed for learners of all ages and backgrounds [4]. This hands-on approach not only improves memorization, but also fosters a lifelong love of learning [5]. Through curated exhibitions, interactive displays, and guided tours, museums have the power to spark curiosity, encourage critical thinking, and deepen understanding. The potential for experiential learning in museum spaces is a cornerstone of their educational importance. Furthermore, the dynamic nature of museums, where exhibits can be replaced or updated, ensures that learning remains a living, evolving process.

One of the primary responsibilities of museums is the preservation of our historical past. They preserve cultural artifacts, documents, and works of art, ensuring their longevity for future generations. This stewardship is essential in perpetuating the memory of the events and individuals that have shaped societies [6]. Memorial museums, such as those

dedicated to sites of trauma and post-communist transitional justice, serve as prime examples of how museums embody the obligation to remember and learn from history [7]. They serve as platforms for negotiating complex historical narratives, shedding light on the challenging process of reckoning with the communist past [8].

In an increasingly interconnected world, the importance of museums in representing diverse cultural heritage cannot be underestimated. They serve as platforms to celebrate cultural pluralism by allowing visitors to learn about the customs, traditions, and worldviews of different communities [9,10]. They serve as cultural ambassadors, showcasing the richness and diversity of a society's traditions, art, and achievements. In this way, museums become important spaces for fostering a sense of belonging and pride among communities.

In addition to their educational and cultural role, museums offer therapeutic benefits that extend to mental health and well-being. For example, art therapy has gained recognition as a powerful tool for healing and self-expression [11]. Art therapy in a museum setting holds promise for relieving stress, anxiety, and trauma, offering a unique way to heal and express oneself [12]. In addition, the peaceful and contemplative environment of museums provides respite for people seeking solace and introspection. Moreover, the inclusive and accessible design of contemporary museums enhances their therapeutic potential by serving a diverse demographic of visitors, including those with special needs [13]. Museums are havens for people facing a variety of physical and mental health challenges. This emerging role of museums as therapeutic spaces highlights their versatility and ability to meet diverse human needs.

Open-air museums occupy a unique niche in the cultural landscape, seamlessly combining the charm of the natural environment with the narrative of human history. Unlike indoor museums, open-air museums present a unique challenge and opportunity for curators. They require innovative approaches to contextualize artifacts in their natural environment [14]. This fusion of nature and history creates an environment in which the past brings the present to life, offering visitors an immersive journey through time and space.

In the digital age, the dynamics of museum experiences are changing. Visitors are no longer looking for a one-size-fits-all approach to museum exhibitions [15]. Instead, they desire immersive, tailored experiences that resonate with their unique interests and preferences. Achieving this level of personalization in a museum environment is a complex challenge. There is a growing realization that offering static exhibitions can limit the potential of museums to engage diverse and increasingly digitally savvy audiences. Museums need to adapt to the expectations of today's visitors, who are used to personalized content across different aspects of their lives, from streaming services and online shopping to social media channels. The realization that museums need to keep pace with these changing visitor expectations has sparked a growing interest in applying artificial intelligence (AI) techniques to provide personalized content delivery [16].

The Center for the Future of Museums [17] identified the personalization of content delivered in museums as one of the six most important trends for 2015. The goal of personalization is to enhance the museum experience for each specific visitor, tailored to their preferences and needs. In recent years, museums have increasingly recognized the importance of delivering personalized content to their visitors [18]. The importance of personalized content delivery in museums extends beyond visitor engagement. It also touches on a critical issue in museum management—how to make collections, exhibitions, and historical narratives relevant and accessible to an increasingly diverse and digitally oriented audience. By harnessing the power of artificial intelligence, museums can analyze visitor preferences and behavior to adapt content, thereby democratizing access to culture and knowledge. In an age where information is abundant and attention spans are limited, personalized content delivery not only captures the visitor's attention but also ensures that the narratives presented are meaningful.

As visitor experiences become increasingly dynamic and technologically driven, the role of artificial intelligence cannot be underestimated. AI offers the unique ability to

process massive amounts of data, learn from visitor interactions, and make real-time recommendations that adapt to the visitor's desires. The use of AI algorithms for personalization has already proven successful in industries ranging from e-commerce platforms offering product recommendations to streaming services offering content [19]. The museum environment presents a unique challenge as it requires AI to consider both the physical layout of exhibitions and the rich historical and cultural context. However, these challenges provide equally unique opportunities for AI to revolutionize the way we interact with cultural heritage.

Although the personalized delivery of content to museum visitors using artificial intelligence is in its infancy, the potential benefits are far-reaching and profound [20]. The purpose of this research paper is to explore the critical importance of advancing personalized content delivery in museums and highlight the key role that artificial intelligence plays in enabling this transformation. As museums continue to adapt to the changing expectations of diverse and digitally minded audiences, the use of artificial intelligence to personalize content offers a promising path to enriching the visitor experience and ensuring that cultural, historical, and artistic treasures in museums remain accessible and relevant in the digital age.

The paper introduces ExhibitXplorer, an innovative service with a distributed architecture that harnesses the power of contextual geofencing, artificial intelligence, and microservices to transform content delivery in the museum context. Particularly important is the proactive content generation methodology driven by real-time analytics of museum visitors' proximity to exhibits. This new approach is revolutionizing the way visitors engage with museum content by offering them personalized experiences that are dynamically tailored to their physical location within museum spaces. ExhibitXplorer delivers content that is not only relevant, but timely. Using artificial intelligence ensures that the content presented to visitors is not only contextually relevant but also intellectually enriching. This paper explores the potential of this service to revolutionize the outdoor museum experience by bridging the realms of history, nature, and cutting-edge technology.

The remainder of this article is organized as follows. Section 2 analyzes related work. Section 3 describes the overall service architecture. Section 4 discusses the validation of the proposed service, and finally, Section 5 concludes the paper and presents some ideas for future work.

## 2. Related Work

The main goal of any museum is to provide information about its exhibits in a way that visitors are satisfied, as well as gain new knowledge in a way that they prefer [21]. In this case, visitors can be expected to visit the museum again after a certain period. The presentation of the exhibits should engage the attention of each visitor for the entire time they spend in the museum. This goal is not easily achievable, as each visitor has specific interests and preferences for how and through which media they receive information.

Personalized content delivery enhances visitor satisfaction by offering customized experiences that resonate with individual preferences. Personalized content delivery allows museums to address the diverse needs of their visitors. Engagement is a critical factor in fostering meaningful connections between visitors and museum collections. Research indicates that personalized content delivery can significantly enhance engagement levels [22]. Personalized content delivery facilitates engagement by aligning exhibits and information with visitors' existing knowledge and interests. By presenting content in a manner that resonates with visitors, museums can ignite curiosity, encourage active exploration, and facilitate deeper connections with the material on display. By utilizing visitor preferences, previous interactions, and demographic information, museums can deliver tailored content, such as interactive exhibits, multimedia presentations, or personalized tours.

Delivering personalized content elevates the overall museum experience by creating a sense of individual connection and relevance [23,24]. The authors of paper [25] present a model for the Museum Exhibition User Experience (MEUX) by combining existing

knowledge about museum visitors' experiences with user experience concepts. The model was developed based on research interviews and surveys with museum professionals in the United Kingdom and presents the museum exhibition experience from both the museum and visitor perspectives.

Personalization can be at the level of a group of visitors with similar interests or for each individual visitor. This requires building as detailed a profile as possible for each visitor. Visitor profiling [26] is the process of collecting and analyzing data about museum visitors to understand their characteristics, interests, preferences, and behaviors. It involves gathering information that helps create visitor profiles, which serve as a foundation for delivering personalized content and experiences. Profiling is most often implemented based on the following information:

- *Demographics*: Visitor profiling includes collecting basic demographic information such as age, nationality, and language preferences. These data help in segmenting visitors and tailoring content to specific groups.
- *Interests and preferences*: Understanding visitors' interests and preferences is crucial for personalization. It involves capturing information about their preferred topics, art styles, historical periods, or specific exhibits they are interested in. These data help curators and the software system to recommend relevant content.
- *Past interactions*: Tracking visitors' past interactions with the museum, including exhibits they have viewed or events attended, provides insights into their engagement levels and preferences. This information helps in refining recommendations and creating a more personalized experience.
- *Visitor feedback*: Collecting feedback from visitors through surveys, feedback forms, or interactive interfaces allows for gathering subjective information about their experience. It helps in understanding visitor satisfaction, areas for improvement, and can provide valuable insights for enhancing personalization efforts.
- *Social media integration*: Integrating with social media platforms allows visitors to connect their social media accounts and share their interests, check-ins, or favorite exhibits. These data can be used to create visitor profiles and provide personalized recommendations based on their social media activity.

Visitor profiling is implemented both implicitly and explicitly [27]. Implicit visitor profiling refers to the collection of user information without their direct input or conscious effort. Implicit profiling involves analyzing visitor behavior and interactions to infer their interests and preferences. This can be conducted by tracking their exhibit preferences, interaction patterns, and engagement levels. Explicit profiling, on the other hand, involves visitors explicitly providing information about their interests and preferences usually during the registration process. This information is used to build their profile and tailor content recommendations accordingly [28].

Explicit profiling mainly uses questionnaires, which can be implemented both at the entrance and exit of the museum [29]. At the current stage, questionnaires are mostly conducted electronically through a mobile application, part of a personalized content delivery application. In addition to demographic data, it is also possible to obtain information about the estimated time of stay in the museum and whether the visitor has any disabilities that would be important for the visitor's navigation and the format of the information delivered.

With implicit profiling, the goal is to dynamically obtain information about visitor preferences. Implicit segmentation involves analyzing visitor behavior, such as patterns of interaction with exhibits. Implicit profiling is imperative as information may not be obtainable from explicit profiling. This means that the personalized content delivery system may not have any information about the given visitors. In this case, the system is in a "cold start" mode [28,30]—it cannot deliver personalized content due to the lack of the visitor's profile. The profile is built over time by tracking visitors' movements, their location, and the time spent viewing the exhibits. Very often profiling is implemented by segmenting visitors—assigning each visitor to one or several specific segments. For example, in [31], the authors define three segments: (a) greedy—visitors with broad interests; (b) busy—

visitors who want brief information about the exhibits; and (c) selective—visitors who want detailed information, but only about specific exhibits. The authors of paper [32] propose to use a classification method called latent class analysis to study museum visitor profiles using a combination of socio-demographic variables and other personal characteristics.

Implicitly profiling visitors requires locating them accurately and involves tracking their movements within the museum. Depending on whether the museum is indoors or open-air, different sensors can be used to localize visitors. The technologies most used are GPS, Wi-Fi access points, Bluetooth Low Energy beacons, and Near Field Communications (NFCs) tags [33]. Museums have the potential to revolutionize their marketing strategies by using location-based techniques such as geofencing. Geofencing allows the creation of a virtual fence around certain exhibits [34]. The proposed application uses both indoor and outdoor geofencing. In indoor museums, BLE beacons [35,36] can be used to form a geofence (circle) in which visitors can detect the presence of a beacon using a mobile app. After entering this protected area, the visitor receives a push notification through which they can confirm or decline receiving additional information about the exhibit associated with the beacon. Similarly, in open-air museums, it is possible to define a protected area, which can be a circle (GPS coordinates of center and radius) or a polygon (GPS coordinates of vertices). In the case of open-air museums, this can include buildings that are exhibits themselves. When visitors walk past them, they can receive a personalized description of the object and so raise their awareness of what is inside. In [35], the authors propose an indoor localization system to enhance the user experience in museums. The system relies on the proximity and localization capabilities of Bluetooth Low Energy (BLE) beacons to automatically provide users with cultural content related to the artworks being observed. While museums are in the early stages of using location-based technologies to increase engagement with their visitors, adoption of geofencing and beacon technology by cultural organizations and museums can be expected in the coming years.

Delivering personalized content can be implemented in several ways, but one of the most suitable is the use of push notifications. Its main advantage is that it delivers messages with up to 90% probability [37]. Used on mobile devices, they capture the attention of customers and are a key strategy for delivering advertising and other personalized content. Click-through rates for push notifications are up to 7 times higher than email marketing. With good target group segmentation, the open rate of push messages jumps by 21%—and personalization can quadruple the open rate of push notifications, suggesting that they are a great tool for increasing engagement.

In 2023, communication and digital technologies are becoming increasingly important for museums and cultural organizations. Expectations of high growth in museum attendance following the pandemic have not fully come to fruition. Some museums have adapted and started to deliver virtual content to their customers using technologies such as artificial intelligence (AI) and virtual reality (VR) [38,39]. The main goal should be to bring visitors back to the museum after COVID-19 [40,41]. However, traditional exhibit presentation, such as text description and audio guides, cannot be relied upon for this purpose. Content must be delivered in such a way that visitor satisfaction and engagement is high. Each museum's content delivery strategy should be based on personalizing the content delivered through accurate visitor profiling. The authors of [42] aim to explore how museums will change after COVID-19. After interviews, they conclude that museums are working hard to achieve sustainable competitiveness after the pandemic, for which they have largely changed their existing business models. One of the most significant changes they introduced was considering their users as internal, rather than external, stakeholders.

Currently, breakthroughs in AI are transforming almost every industry. Museums are not left out. AI is being used to create systems to help curate exhibits, reliably archive and store them, and improve the visitor experience using chatbots, robot interpreters, and multilingual translators [43,44]. According to cuseum.com, nearly 80% of museums view new technology as a critical success factor in attracting visitors, improving content relevance, and diversifying audiences. Silicon Valley startup OpenAI has developed a

generative artificial intelligence known as the Generative Pre-trained Transformer (GPT) and the DALL-E model, a digital art generator powered by artificial intelligence [45]. Museums can use ChatGPT to deliver personalized and engaging content to increase their visitor numbers as well as improve their engagement and satisfaction. ChatGPT can provide language support through its chatbot service. Visitors to the museum will be able to communicate in their preferred language, improving their overall experience at the museum. The authors of the paper [46] present a novel approach to develop creative artificial intelligence by co-creating language models using Open AI GPT-3. The authors use the four measures of creativity: fluency, flexibility, elaboration, and originality to generate creative behaviors. The analysis of the resulting model was tested through storytelling for children.

As technology evolves, more museums are exploring cloud platforms to securely store and manage their growing collections. Providers such as Microsoft Azure, Amazon Web Services, and Google Cloud Services offer cloud services at relatively affordable prices through discounts for nonprofits. Museums have more options than ever to reliably store and manage their digital assets in the cloud. Taking all of this into account, the European Commission has initiated a dialogue in 2022 on the co-creation of a cultural heritage cooperation cloud [47]. The aim is to support the preservation of cultural heritage in the European Union through a common digital infrastructure. The cloud will enable the digitization of exhibits and works of art. The aim is to give access to the latest technologies to smaller institutions that have limited budgets.

The main objective of this article is to present the architecture, implementation details, and evaluation of a service for delivering personalized content to museum visitors called ExhibitXplorer. The effectiveness of the system for delivering personalized content is evaluated. The results of surveys and experiments show that the quality of content generated and overall user satisfaction with the system are at a high level.

## 3. System Architecture

To address the challenges of developing services to deliver personalized content to museums, the ExhibitXplorer service was designed. It has a distributed architecture that uses geofencing, artificial intelligence, and microservices. ExhibitXplorer aims to transform the museum experience by seamlessly integrating cutting-edge technologies and intelligent systems. By combining implicit and explicit visitor segmentation techniques, the system can tailor content to the unique interests and preferences of different visitor segments. Implicit segmentation involves analyzing visitor behavior, such as interaction with previous exhibits, to infer their interests. Explicit segmentation enables visitors to self-identify their profiles by selecting predefined categories that best represent their characteristics, such as researcher, student, casual visitor, or child. The integration of these segmentation methods allows for a comprehensive understanding of visitors' preferences and ensures accurate content customization.

The personalization is based on contextual geofencing [48] and artificial intelligence. The application is proactive towards content delivery. Personalized content starts to be generated when the visitor approaches an exhibit of interest. The artificial intelligence is a language model trained to index the textual description related to the museum and its exhibits. Through it, it is possible to automatically generate the description of the exhibits in the database, as well as implement a chatbot to deliver personalized and language-independent content. To use the app, all that is needed is a mobile device with the Android operating system and an installed mobile app to communicate with the business logic.

To deliver personalized content, ExhibitXplorer harnesses the power of the GPT, a language model developed by OpenAI. The authors of the paper [49] compare multiple state-of-the-art conversational models such as ChatGPT, Galactica and KGQAN and conclude that ChatGPT has the best capability in terms of explainability and robustness. Rudolph et al. [50] compared ChatGPT, Bing Chat, Bard, and Ernie chatbots and concluded that GPT-4 and its predecessor performed best on a multidisciplinary test related to higher

education. GPI API enables real-time content generation based on the type of segment the visitor belongs to. The utilization of the GPT API ensures that the information delivered is fresh, relevant, and aligned with the specific needs and interests of each visitor segment.

Furthermore, ExhibitXplorer employs push notifications as the primary means of delivering personalized content to museum visitors. These push notifications are triggered when visitors approach specific exhibits or enter predefined geofences. By leveraging geofencing, the system ensures accurate and timely content delivery. The potential impact of ExhibitXplorer extends beyond enhancing museum visits. It sets a precedent for the integration of technology and personalization in cultural and educational institutions, paving the way for innovative approaches to engage visitors, foster learning, and promote a deeper understanding of our shared heritage and knowledge.

The proposed ExhibitXplorer software system has a distributed architecture that is based on microservices. Compared to centralized applications, a distributed architecture offers several advantages, including scalability, fault tolerance, and flexibility. This architecture is particularly suitable for delivering personalized content as it allows for modular development, easy integration of various services, and efficient handling of visitor segmentation and localization. The distributed architecture of the ExhibitXplorer system enables efficient communication between microservices, seamless integration of various components, and scalability to handle many visitors. It leverages the strengths of Node.js and Python for different microservices, utilizing message brokers and databases to facilitate communication and data storage.

Figure 1 shows a block diagram of the application architecture. The ExhibitXplorer system has been implemented using a combination of technologies and frameworks to ensure efficient and reliable operation. The technology stack in ExhibitXplorer was carefully selected to provide a robust, scalable, and efficient system that delivers personalized content to museum visitors, enhancing their overall experience and engagement.

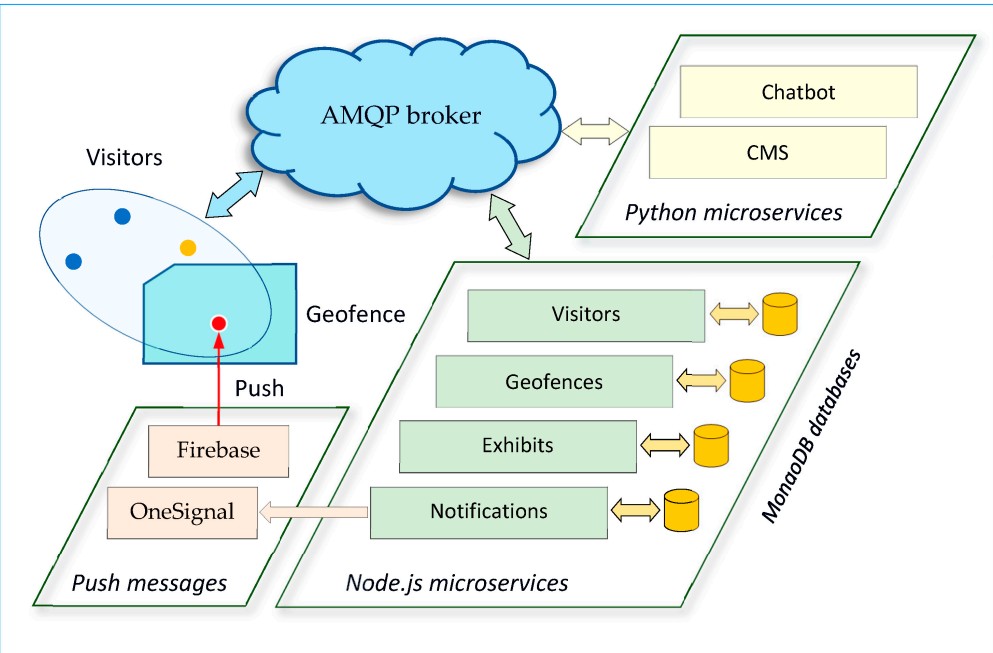

**Figure 1.** ExhibitXplorer architecture.

The technology choices in ExhibitXplorer were carefully considered to optimize performance, efficiency, and user experience. Node.js was selected for its event-driven and non-blocking I/O model, which allows for handling concurrent requests efficiently. Python, with its rich libraries and frameworks, was chosen for the chatbot and CMS microservices. MongoDB, a NoSQL document-oriented database, was selected for its flexibility and scala-

bility. MongoDB's geospatial indexing and querying capabilities make it a suitable choice for the geofences microservice, enabling fast geofence-based content delivery.

According to the American Alliance of Museums [51], one of the trends for 2023 is the use of Web 3.0 in museums. Web 3.0 is a term that describes a decentralized Internet built on technologies such as blockchains that allow data storage and control to be distributed across servers owned by many individuals or organizations. The Interplanetary File System (IPFS) is a distributed file system used in this development. Infura IPFS was chosen for multimedia file storage due to its decentralized and distributed nature. IPFS ensures content availability and reliability, and the content identifiers (CIDs) and MIME type of each multimedia file were recorded in the MongoDB database for easy retrieval.

The adoption of a distributed architecture in ExhibitXplorer offers several advantages over a centralized approach. Firstly, it allows for better scalability and load distribution. By dividing the system into microservices, each responsible for a specific business logic, the system can handle many concurrent visitors and requests. This ensures a seamless and responsive user experience even during peak periods. Secondly, a distributed architecture enhances fault tolerance. In a centralized system, a single point of failure can disrupt the entire system. In contrast, in ExhibitXplorer, if a microservice fails, the rest of the system can continue to function independently. This fault isolation ensures that the system remains operational, reducing downtime, and improving reliability. Finally, a distributed architecture enables flexibility and modularity in system development. Each microservice can be developed, deployed, and updated independently, allowing for agile development practices. This flexibility facilitates easy integration of new functionalities and technologies, ensuring the system remains adaptable to evolving requirements.

For the *visitors* microservice, Node.js provides a non-blocking and event-driven architecture, enabling efficient handling of concurrent visitor requests and communication with other components. The microservice communicates with other services through a message broker, RabbitMQ, ensuring reliable message delivery and decoupling of components. MongoDB is employed as the database for its flexibility and scalability, allowing seamless storage and retrieval of visitor data. The *exhibits* microservice manages access to data related to exhibits in the museum. This microservice uses the MongoDB database for the description of exhibits. All exhibit properties are initialized through the interface that the CMS microservice provides. The initialization of the values of these properties can be implemented automatically by using the content generation capabilities through the GPT API. The *geofences* microservice leverages the capabilities of MongoDB for efficient geospatial queries. MongoDB's geospatial indexing and querying features enable fast and accurate determination of a visitor's proximity to geofences. This microservice plays a crucial role in providing real-time geofence-based content delivery to visitors. The *CMS* microservice uses the FastAPI framework, known for its high performance and scalability. FastAPI offers asynchronous capabilities, enabling efficient handling of multiple exhibit requests and providing a responsive Web interface. MongoDB is employed for exhibit data storage, while multimedia files are stored in Infura IPFS. IPFS ensures decentralized and reliable storage of multimedia files, with the MongoDB database recording the necessary information for their retrieval. The *chatbot* microservice is used as an interface to the GPT API. It enables the system to generate personalized content based on visitor profiles. By leveraging the GPT API, the system can provide tailored responses to visitor queries, enhancing the interactive and informative nature of the museum experience. The notifications microservice uses the OneSignal service for push notification delivery. OneSignal offers a user-friendly interface and robust notification delivery capabilities. By integrating this microservice into the system, personalized push notifications can be sent to specific visitors or visitor groups, ensuring timely and relevant information delivery.

### 3.1. Visitors Microservice

The visitors microservice handles the registration, segmentation, and localization of visitors. Communication with this microservice is implemented through a message broker that provides reliable and asynchronous communication between components. MongoDB is used as the database, offering flexibility and scalability.

Visitor segmentation in ExhibitXplorer plays a crucial role in personalizing the content delivery. Visitor profiling in museums can provide valuable insights into visitor preferences, behaviors, and interests, enabling museums to personalize experiences, tailor exhibits, and improve overall visitor satisfaction.

In this development, hybrid visitor profiling is used (combining explicit and implicit profiling). Explicit profiling involves directly collecting information from visitors using short survey forms (one to three questions) that are sent to visitors after analyzing their activity. The aim is to provide demographic data such as age group, education, occupation, nationality, etc. Explicit profiling also involves capturing visitors' preferences, motivations, and expectations regarding their museum experience. It also provides information about the amount of time the visitor intends to spend in the museum and the presence of any disability. Requests for explicit profiling are sent as push notifications. The visitor can refuse to answer one or all questions. Implicit profiling, on the other hand, involves analyzing visitor behavior and their interactions with exhibits. In ExhibitXplorer, this method uses various techniques to observe and collect data about visitor actions. Using geofencing (BLE beacons and GPS), the visitors' movements within the museum are tracked. Visitor engagement with the exhibits is analyzed by recording the time spent viewing the exhibits (dwell time). Visitors are also segmented based on the frequency of their visits to the museum. Understanding patterns of repeat visits can help museums develop loyalty programs, personalized recommendations, or targeted promotions to encourage repeat visits.

At this stage, the following visitor segmentation approaches are used:

- *Demographic segmentation*: Age Groups—(children and teenagers, adults, and seniors); Education Level (high school diploma, undergraduate degree, postgraduate degree, or professional qualification); Occupation (students, teachers, artists, engineers, business executives, etc.); and nationality.
- *Behavioral segmentation*: Exhibit Preferences (enthusiasts, history buffs, science lovers, technology enthusiasts, or nature enthusiasts); Dwell Time (quick explorers, average visitors, or avid explorers who spend extended periods at exhibits); Repeat Visits (first-time visitors, occasional visitors, or frequent visitors).
- *Motivational segmentation*: Personal Interest Motivation (art, history, science, technology, nature, etc.); Inspirational Motivation (artists, designers, innovative ideas, etc.).
- *Experience-based segmentation*: Expert Visitors; Art Enthusiasts; Cultural Heritage Seekers.

The number of segments in visitor segmentation can affect the ability to deliver personalized content. A larger number of segments allows for more precise targeting and tailoring of content to specific visitor preferences, interests, or behaviors. It enables museums to provide highly personalized experiences that resonate with individual visitors. However, managing many segments requires significant resources, including data collection, analysis, and content creation.

### 3.2. Geofences Microservice

The geofences microservice provides an interface to access the MongoDB database, which contains all geofences. At this stage, two types of geofences are supported: a circle with a specified center and radius and a polygon described in the GeoJSON format. All beacons associated with exhibits form a geofence that is modeled by a circle. Outdoor exhibits are described by geofences in polygon form. Two applications have been developed to build the database of outdoor geofences: (1) A web application that allows the input of geofences using GPS maps. The application allows the entering, editing, deleting, and exporting of geofences. The export can be to a GeoJSON file or directly to the database.

(2) A mobile application with similar functionality but with the ability to display the current position of the client. The goal is to provide the ability to enter the coordinates of geofences outdoors that are missing on GPS maps—for example, remains of historical landmarks. The application exports the entered geofences directly to the database.

MongoDB was chosen as the database because it has very good capabilities for working with geospatial data. The database provides specific geospatial queries that make it possible to check in real time which visitor is near to which geofences or in which geofences at a given time. It is also possible to obtain information about which visitors are in each geofence. The response time for geospatial queries is several milliseconds for tens of thousands of visitors and geofences. The reason for this is that MongoDB does not actually work with GPS coordinates but describes the contour of each geofence using geohashes. A geohash is a string of a certain length that is a unique identifier of a specific geographic region with a rectangular shape. The longer the string, the more accurately the region is described.

The proposed service uses contextual geofencing to deliver personalized content. Contextual geofencing refers to the use of location-based technology to define virtual boundaries or geofences around specific physical locations. These geofences can trigger certain actions or deliver targeted content when a user's mobile device enters or exits the designated area. In contextual geofencing, both static and dynamic contexts are used to define and enhance the location-based experiences. The static context includes geographic coordinates of the geofence; an exhibit associated with the geofence; and static content associated with the exhibit, such as multimedia content. Dynamic context includes hours of operation; date and time; day of week; visitor profile; visitor's moving mode (still, walking, running, and transportation); and weather conditions. By combining these static and dynamic contexts, a contextual geoengineering system can analyze a visitor's current location and moving mode, weather, visitor's preferences, and movement to provide relevant and personalized content. The system can dynamically adjust content based on real-time information and visitor interaction, creating a more engaging and enjoyable experience for museum visitors.

Contextual geofencing can be used to segment museum visitors. By implementing geofencing technology, museums can gather valuable data about visitor movements, behaviors, and interactions within specific areas of the museum. These data can then be used to create visitor segments based on location-based preferences and activities. The context most associated with geofencing in the museum setting is the physical location within the museum premises. Geofencing can help museums understand how visitors move through different exhibition spaces and how visitors engage with specific exhibits within the museum. This contextual information can be used to personalize content, provide targeted recommendations, and optimize the overall visitor experience. Geofencing can also be used to enhance engagement by triggering location-specific notifications, such as providing additional information or multimedia content when visitors approach certain exhibits or points of interest. For example, when a visitor enters a geofenced area around a particular artwork, their mobile device could display detailed information about the artwork, artist, or related events. Furthermore, geofencing can be used to gather data on visitor flow, crowd density, and queue management. This information can assist museums in improving exhibit layouts, optimizing resource allocation, and enhancing visitor circulation within the museum. It is not enough for a visitor to be in the vicinity of a geofence to generate a personalized response. The dynamic context associated with the geofence must also be analyzed.

While geofencing provides valuable contextual information, it is important to implement it responsibly, respecting visitor privacy, and obtaining appropriate consent or permissions. Clear communication and transparency regarding data collection and usage are essential when utilizing geofencing technology. Overall, contextual geofencing offers museums an opportunity to gain insights into visitor behavior, personalize experiences, and optimize operations based on location-based data within the museum environment.

### 3.3. Exhibits Microservice

The exhibits microservice provides access to a database that stores information about each exhibit in the museum. Entering information into this database is implemented using a CMS microservice that provides a Web interface to the museum curators. The textual information is stored in a MongoDB database and the multimedia files in the IPFS distributed file system. For each multimedia file (images, audio, and video), the IPFS file content identifier (CID) and the tags associated with it are recorded in a MongoDB database. Setting tags for each content type is necessary to implement its filtering when delivering personalized content more easily.

The exhibits microservice provides interface methods through which non-personalized or personalized content can be obtained for a requested exhibit. Communication with this microservice is via a message broker using the Advanced Message Queuing Protocol (AMQP). When delivering non-personalized content, the microservice retrieves information from the database and returns it to visitors who are not yet profiled (cold start). If a profile has been created for the visitor, the microservice attempts to return personalized content. This is implemented in two possible scenarios. If the chatbot microservice is functional, the personalization is implemented via ChatGPT. If for some reason the chatbot microservice does not function, the personalization is implemented based on filtering the database content by considering the tags for each media type.

### 3.4. CMS Microservice

The CMS microservice is implemented in Python. It uses the FastAPI framework to create the Web user interfaces. FastAPI was preferred over Flask or Django for the CMS microservice due to its exceptional performance and scalability. Its asynchronous capabilities enable efficient handling of exhibit requests, ensuring a responsive web interface for users. This microservice allows creating and editing information in the database for each exhibit.

Figure 2 shows the functioning of the CMS microservice. CMS communicates with exhibits and chatbot microservices via a queue named CMS. The microservice provides a REST interface with multiple endpoints through which information about a selected exhibit can be created, edited, or deleted. The textual information about the exhibit (name, description, author, year of creation, material used, dimensions, location in the museum, etc.) is recorded in a MongoDB database using the exhibits microservice.

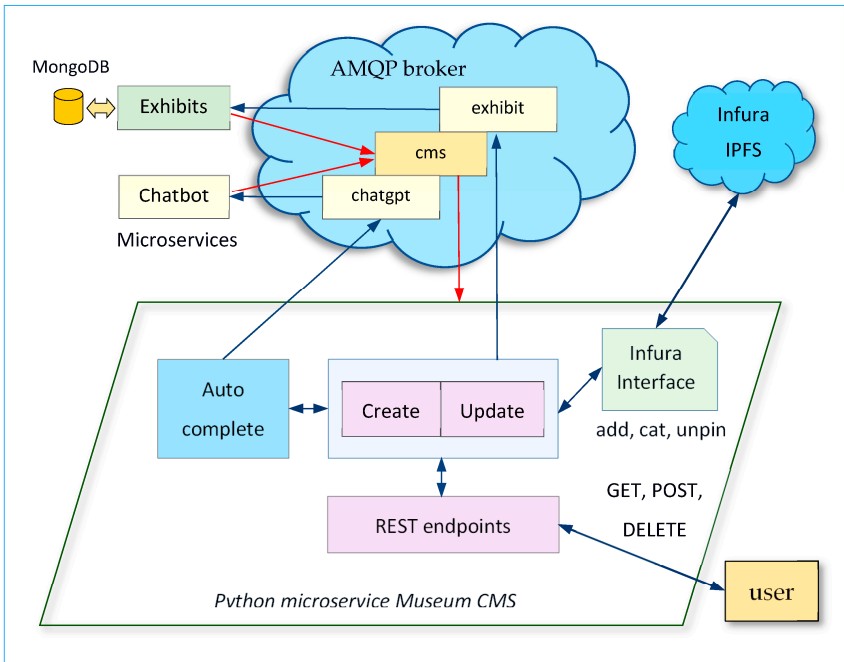

**Figure 2.** CMS microservice.

An HTML form is used to describe the exhibits. The only required fields are the name of the exhibit and its detailed description. Filling in the remaining fields of the form can be implemented automatically via ChatGPT. For this purpose, appropriate prompts are generated through which ChatGPT generates the missing information. For this purpose, it is necessary to pass the complete description of the exhibit as the value of the assistant field of the GPT query. All requests are sent to the chatbot microservice, which implements the requests to ChatGPT.

Multimedia files, including images, videos, and audio, can be associated with each exhibit. The multimedia files are written to the IPFS distributed file system. IPFS is ideal for this purpose as it provides a decentralized and distributed file system, ensuring content availability and reliability. The MongoDB database records only file CIDs and MIME type of each multimedia file, facilitating their retrieval.

### 3.5. Chatbot Microservice

The chatbot microservice, implemented in Python, enables query submission to the ChatGPT using the GPT API. This microservice retrieves personalized information based on visitor segmentation, ensuring customized responses to inquiries and questions.

Museum visitors' queries to the chatbot microservice are implemented through a message broker, as Figure 3 shows. A microservice chatbot works as a server that implements visitor requests. Using a mobile application, visitors send their requests. They are transmitted to a ChatGPT queue. The requests are JSON objects that contain the following information:

- *visitor_id*—visitor identification string.
- *timestamp*—specific request time in Unix format.
- *command*—name of the command that is passed to the microservice. Two command names are recognized: (1) prompt (user request) and (2) reset—clears the context associated with the given visitor.
- *system*, *assistant*, and *user*—content of the request.

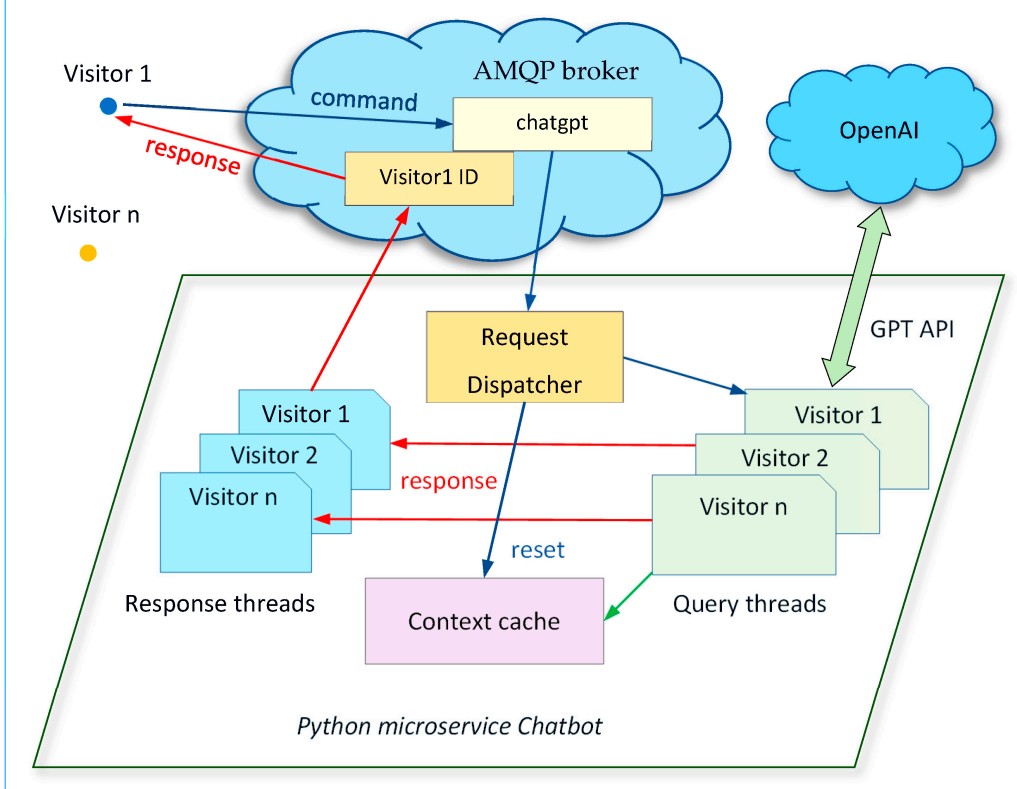

**Figure 3.** Chatbot microservice.

The "user" field contains the visitor request itself. In the context of the GPT API, the "system" and "assistant" fields are used to provide additional instructions and context to the language model, allowing you to customize and guide the behavior of the model. The "system" field is used to specify the initial behavior or role of the assistant. It helps to establish the context of the conversation. The "assistant" field is used to track the context of the conversation. It records both the questions and the chatbot's responses. When communicating between visitors and a chatbot microservice, the "system" and "assistant" fields are left blank. This means that the server side must initialize them. In this case, the role being set is museum curator, and the assistant field is initially initialized with the full text information for the specific exhibit.

Through the effective use of the "system" and "assistant" fields, dynamic and interactive conversations are created with the GPT language model. To generate content specific to each visitor, the chatbot microservice adds information to a "user" field so that the visitor's profile is accounted for. Thus, depending on the segments to which it belongs, content unique to each visitor is generated. The following is a sample response that is generated when a visitor who is visually impaired requests a description of the Mona Lisa painting:

> *"The Mona Lisa painting is a portrait of a woman with an enigmatic smile. She is looking directly at the viewer with her eyes, and her hands are crossed in her lap. She is wearing a draped garment and a scarf around her neck. Her facial features and expression are painted with delicate brushstrokes and subtle shading, giving her a mysterious and enigmatic look. The Mona Lisa painting is a portrait of a woman with a mysterious expression on her face. She is seated against a background of rivers and rolling hills. Although the portrait is painted in earth tones, there are hints of yellow and ochre in her face and clothing. Her hands are crossed delicately in her lap, and she stares out at us with her small lips and captivating eyes. The painting is thought to have been painted by the Italian artist Leonardo da Vinci in the 1500s. The size of the painting itself is 77 cm × 53 cm. The painting has become world famous and is admired for its enigmatic expression."*

The description of the picture for a visitor who is a student is as follows:

> *"The Mona Lisa is a famous painting created by the artist Leonardo da Vinci. It shows a woman with a mysterious smile on her face. She is wearing a dark dress and a beautiful headscarf. The painting is very special because it is one of the most famous and recognizable works of art ever made. People come from all around the world to see it!"*

To complete the user requests, each request is processed in a separate program thread. Similarly, in separate program threads, the received responses are processed. The microservice chatbot sends the response to the corresponding visitor using the message broker. The response is sent to a queue whose name matches the visitor ID (see Figure 3).

The microservice keeps the context of the conversation with each visitor through a "context cache" block. This is a synchronized object that can only be accessed by one program thread at a time. Deleting the context is possible via a "reset" command. This command is generated at the start of each new conversation or after a timeout.

Proactive Content Generation

The developed service is proactive in generating personalized content. The response time of the GPT API depends on the pattern set at the request but is no shorter than 5–10 s. For this reason, this content starts to be generated earlier in time—when the visitor approaches an exhibit at a set distance. This distance is specific to each exhibit and is set in the exhibits database. When the visitor gets close to an exhibit, he receives a push notification informing him about which exhibit it is. At this point, the visitor receives only short, non-personalized content. The visitor can confirm or reject the delivery of the personalized content. If the visitor is hesitant, he/she has the option to use the chatbot to obtain further information.

Figure 4 shows the flowchart of the algorithm for proactively obtaining personalized content. It is likely that the content to be generated has already been generated for another visitor with a similar profile. For this reason, before a request is sent to ChatGPT, it is checked that similar content has not already been generated. To this end, the visitors database is searched for visitors who had a similar profile to the visitor $Vn$ for whom custom content is to be generated for exhibit $Ek$. Jaccard similarity is used to check the similarity of visitor profiles. The Jaccard similarity measures the similarity between two sets of data to see which members are shared and distinct. It is calculated by dividing the number of observations in both sets by the number of observations in either set:

$$Sim_{n,m} = (Sn \cap Sm)/(Sn \cup Sm)$$

where $Si$ is the set containing the segments a visitor with identifier $i$ belongs to, and $Sim_{n,m}$ is the probability that visitor $n$ and visitor $m$ have similar profiling. The similarity coefficient is a number in the interval [0, 1]. To quickly calculate the probability that each of the other currently active visitors has a similar segmentation as visitor $Vn$, the database capabilities of MongoDB are used (operators $setIntersection, $setUnion, and $devide).

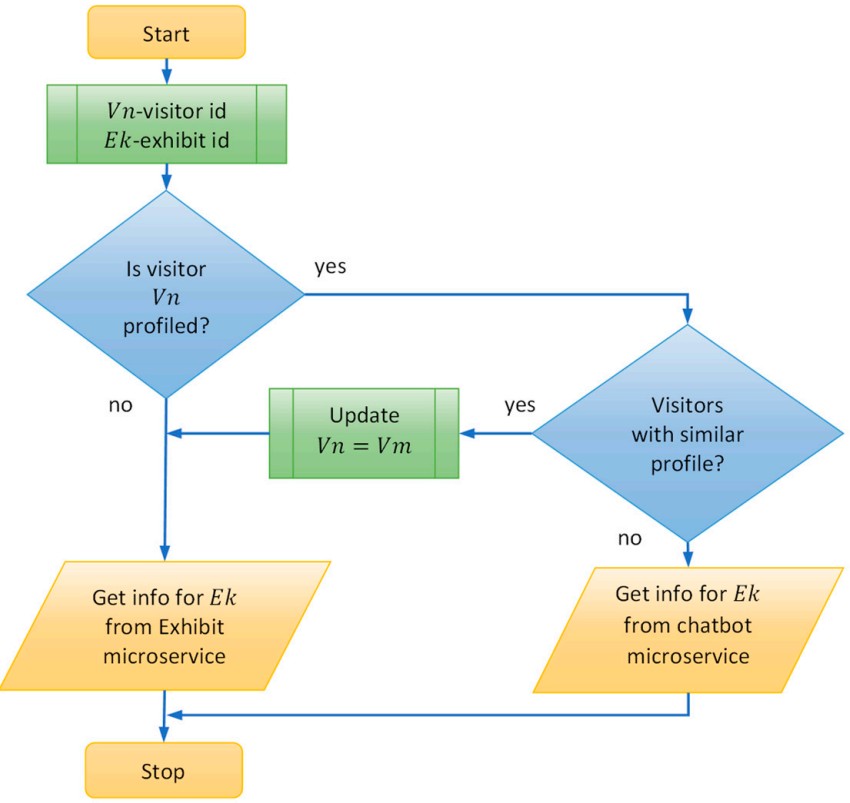

**Figure 4.** Flowchart of the algorithm for proactively obtaining personalized content.

For each visitor $Vm$ with a value for $Sim_{n,m}$ greater than or equal to a predefined threshold $SimTh$, it is checked whether content has been generated for exhibit $Ek$. If such content is missing, the ChatGPT microservice needs to be used to generate custom content for exhibit $Ek$ while accounting for the $Sn$ segments to which visitor $Vn$ belongs. The text part of the response is obtained using the GPT API. For this purpose, the full description of the exhibit $Ek$ is passed as a prompt, and depending on the segmentation, filters are set for this content. The remaining information (images, video, audio, and links to Internet sources) is filtered according to the segments to which the visitor belongs. If the required content has already been generated, for example, for the visitor $Vm$, then this content is returned as a response using the exhibits service.

### 3.6. Notifications Microservice

The notifications microservice, implemented in Node.js, allows the delivery of push notifications via the OneSignal service. OneSignal provides a push notification delivery platform that allows targeted notifications to be sent to individual visitors or specific groups of visitors. OneSignal uses the Google Firebase Cloud Messaging (FCM) service to deliver the push notifications to museum visitors.

Museum visitors receive push notifications when they approach or enter geofence boundaries. ExhibitXplorer uses push notifications as the primary means of delivering personalized content to museum visitors. If a visitor opens a notification associated with an exhibit, the service delivers personalized content for the exhibit the visitor is in proximity to. Push notifications are also used to implement explicit profiling, deliver museum-related promotional content (e.g., new exhibits, ticket price reduction), and when an evacuation plan is activated.

## 4. Results

The proposed service for delivering personalized content has been tested in an open-air ethnographic museum. The workshops in the museum are described as geofences using the GPS coordinates of their contours. Visitors to the museum received personalized content for each of the 36 sites in the museum using a specially developed mobile application.

### 4.1. Software Deployment

Containers are created for all microservices and deployed to Docker Hub. The goal is to transfer the containers to a cloud platform so that the microservices are publicly available. The process of deploying, managing, and scaling container applications is implemented using Kubernetes, a container orchestration platform that automates the deployment, scaling, and management of applications in containers. There are several cloud platforms that can be used to deploy microservices written in Node.js and Python, for example:

- Amazon Web Services (AWS): AWS offers services like Amazon Elastic Container Service for Kubernetes (Amazon EKS) for managing Kubernetes clusters, as well as Amazon Elastic Container Service (Amazon ECS) for container orchestration.
- Microsoft Azure: Azure provides Azure Kubernetes Service (AKS) for managing Kubernetes clusters, allowing to deploy and scale microservices easily. It also offers Azure Container Instances (ACI) and Azure Service Fabric as alternatives for container orchestration.
- Google Cloud Platform (GCP): GCP offers Google Kubernetes Engine (GKE) for managing Kubernetes clusters. It allows us to deploy and manage microservices as containers.
- IBM Cloud: IBM Cloud offers Kubernetes Service for managing Kubernetes clusters, enabling us to deploy and scale microservices.

### 4.2. Databases

For the application to deliver content, it is necessary to create the databases that describe the geofences and exhibits.

#### 4.2.1. Geofences Database

A Web application has been developed that allows a quick description of the contours of the geofences as a polygon or circle with a given center and radius. The possibility of converting all geofences described with a circle to a polygon is provided. Each geofence is assigned a unique name that is used as the geofence identifier in the database. At any time, geofences can be exported to a file or saved to a MongoDB database in GeoJSON format. The application also allows the description of geofences using geohashes to reduce the execution time of geospatial queries. When working with MongoDB and indexing the database with the 2dsphere index, it is not necessary to use a geohash description, as MongoDB automatically implements the conversion of polygons to geohashes.

Figure 5 shows the interface of the application that creates the geofences database. Every visitor to the museum falls into one or more geofences. To keep track of which visitors are in the museum, the outline of the museum itself is a geofence. Several geofences can be combined into a common geofence, e.g., the workshops of the craftsman trade street. A Web app has been developed that can be used to check if the geofences database has been successfully built. The application displays on a GPS map all geofences as well as the position of all visitors if there is already information in the visitor database (see Figure 6).

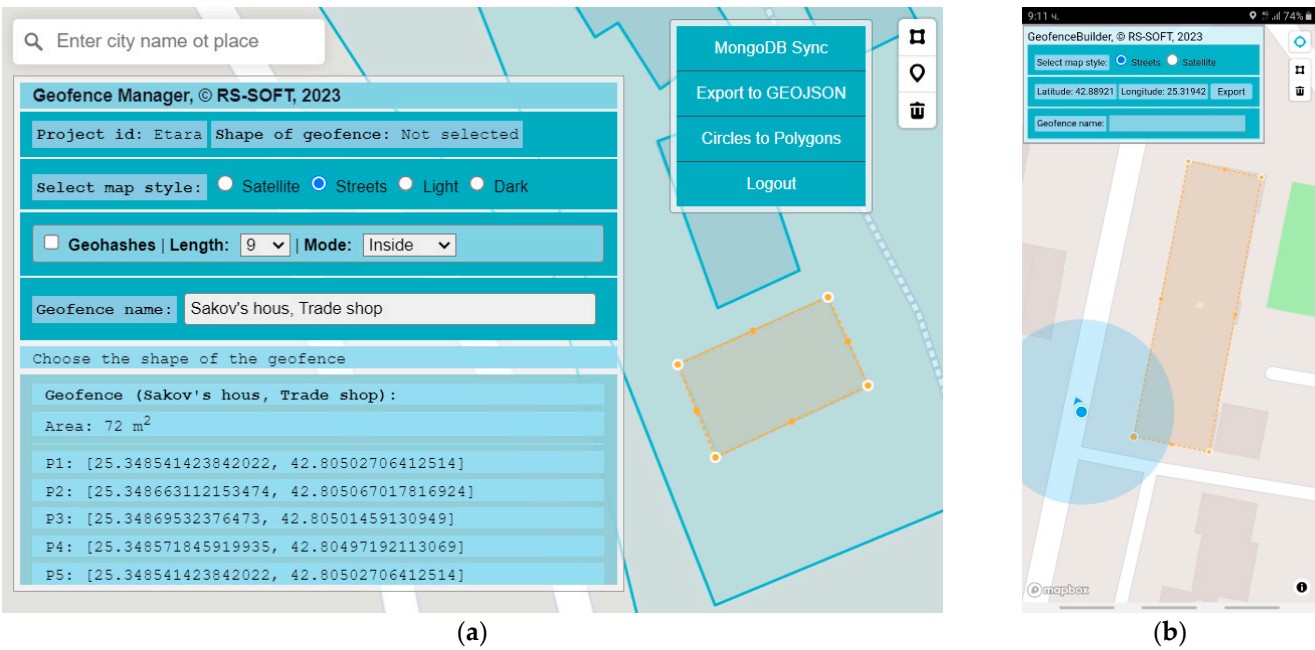

(**a**)  (**b**)

**Figure 5.** Application for creating the geofencing database: (**a**) Web app; (**b**) Mobile app.

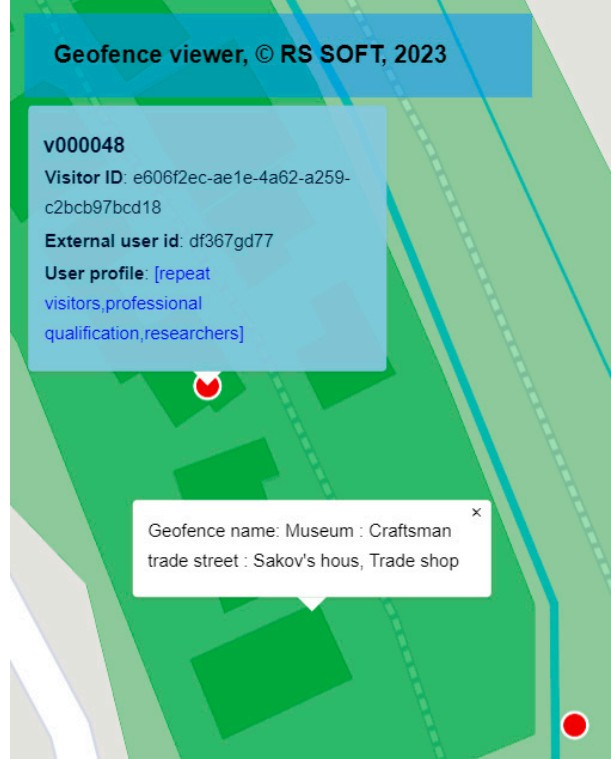

**Figure 6.** Web application for testing the geofences database.

### 4.2.2. Exhibits Database

In open-air museums, each geofence describes a building (e.g., workshop) or an open-air exhibit. In this example, the exhibits are the workshops and shops to which visitors have access. A database of exhibits is created through a CMS microservice. It provides a Web interface through which museum staff can describe exhibits, edit their properties, and delete information associated with an exhibit. For example, when entering a new exhibit, a form is displayed for the staff member to fill out (see Figure 7).

**Figure 7.** Exhibit description form.

Fields in black must be initialized by the user, and those in gray can be initialized automatically. In this example, the information to be entered manually is the exhibit name, the exhibit type, and the media files to be associated with the exhibit. For each selected media file, tags must be entered that are associated with the file. These tags are needed when filtering the multimedia content to return personalized content. The rest of the information can be obtained automatically using the chatbot microservice.

For museums with a very large number of exhibits, the possibility to train their own language model or fine-tuning an existing pre-trained model is provided. This is also necessary in cases where the museum is small and the language model does not have

indexed information. For these museums, an additional microservice with a Web interface is provided that allows museums to easily train their own language model. The training of the model is implemented using the LlamaIndex data framework. LlamaIndex is a data framework designed for creating Large Language Model (LLM) applications. LLMs are pre-trained on publicly available data. LlamaIndex provides a framework for augmenting LLMs with private user data. The available API allows LlamaIndex to be used for data ingestion and search. The framework offers connectors for training data using different file formats, for example, PDF and DOCX. It provides different ways to index the data so that these data can be easily used with LLM and an advanced interface to query the indexed data. At this stage, a gpt-3.5-turbo-0613 model is used as an LLM. This model was chosen because of its good speed performance and the possibility of testing it without any latency. Input data should be prepared by museum curators in PDF or DOCX format. This can be unstructured data as well as questions and answers. Queries are executed using the ChatGPT model when the resulting LLM index file is set as the source.

For museums with few exhibits, another strategy is possible. In this strategy, the detailed description of the exhibit is entered manually, and the rest of the form fields are automatically populated using appropriate ChatGPT prompts based on the detailed exhibit information. For example, if the author of the exhibit is to be retrieved from the exhibit description, the prompt should be, "Return only the name of the exhibit's author." In this way, the data for all exhibits does not need to be indexed, but the functionality of the service is lower since ChatGPT cannot return responses that use information for multiple exhibits.

*4.3. Mobile App*

To access the service ExhibitXplorer, the visitor must install a mobile application for Android OS on their mobile device. The installation of the application is realized automatically by scanning a QR code that is positioned at the museum entrances. After the app is launched for first time, it registers to receive push notifications via Google FCM. Upon successful registration, an FCM access ID is returned. This unique string is also used as the visitor identifier. The ExhibitXplorer service does not require registration by name or e-mail address to ensure the anonymity of visitor data collected.

The mobile app registers listeners for events from NFC, BLE beacons, and GPS. These listeners allow activation of a program code when certain events occur; for example, the mobile device is near an NFC tag; the mobile device is in proximity to a BLE beacon; or new GPS coordinates are received. In terms of battery consumption of the mobile device, the critical hardware part is the GPS receiver. For this reason, the GPS receiver is only accessed when it detects that the visitor is on the move. For this purpose, a human activity detector (HAR) is used to classify the visitor's activity into one of the following categories: still, walking, running, and transportation. Data from BLE beacons and GPS are only analyzed if the HAR returns a walking category. The implementation of HAR is based on the analysis of accelerometer data.

To use the ExhibitXplorer service, the app does not need to be in focus as event listeners from BLE beacons and GPS are running in the background. When a visitor approaches an exhibit, the geofences microservice (via the notifications microservice) sends a push notification. It notifies the visitor that they are near an exhibit (see Figure 8).

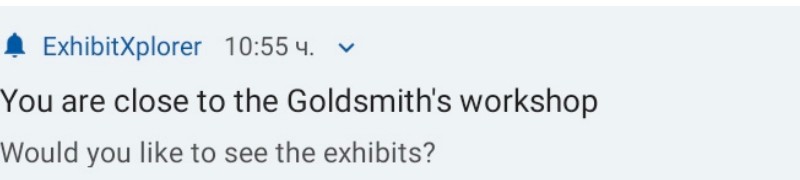

**Figure 8.** Push notification.

The visitor can open the notification or leave it without consequence. If the visitor opens the notification, the service returns only brief information about the exhibit, but begins proactively generating a personalized content. From the brief information, the visitor can find out exactly what the exhibit is, as well as use the chatbot to obtain more information. If the visitor chooses to receive additional information, the service returns the already prepared personalized content.

When the notification is opened, a short content for the exhibit is generated. This content includes the name of the exhibit, a short text description, and an image. If the exhibit is of interest to the visitor, they can receive answers to their specific questions via the museum's chatbot as well as request a full description of the exhibit depending on their current profile (see Figure 9).

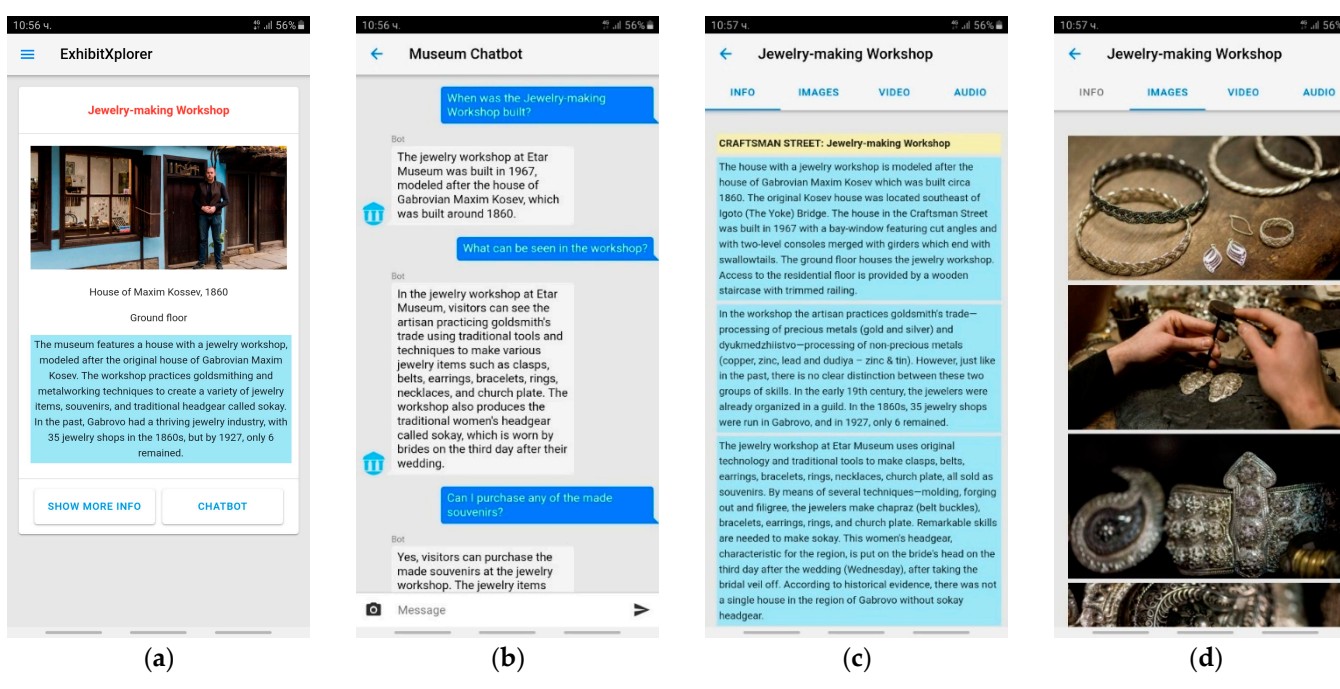

**Figure 9.** Description of the exhibit: (**a**) Brief description; (**b**) Chatbot; (**c**) Full description (text); and (**d**) Full description (images).

The full description contains the types of information that the visitor prefers to receive. In this example, this is text, images, video, and audio files related to the exhibit. References to additional resources and links to the Internet sites are missing.

The mobile app is also used to implement explicit visitor profiling. For this purpose, they receive push notifications inviting them to answer one or more questions. Initially, it is checked how much time each visitor can spend browsing the exhibits in the museum. Depending on the answer, the profiling module uses different strategies. If the visitor declares little time to spend in the museum, a survey form is sent to the visitor that contains all the questions related to explicit profiling. Otherwise, to make the visitor more likely to respond, the questions are sent in parts, each containing one to three questions. Figure 10 shows some of the questions sent to a museum visitor.

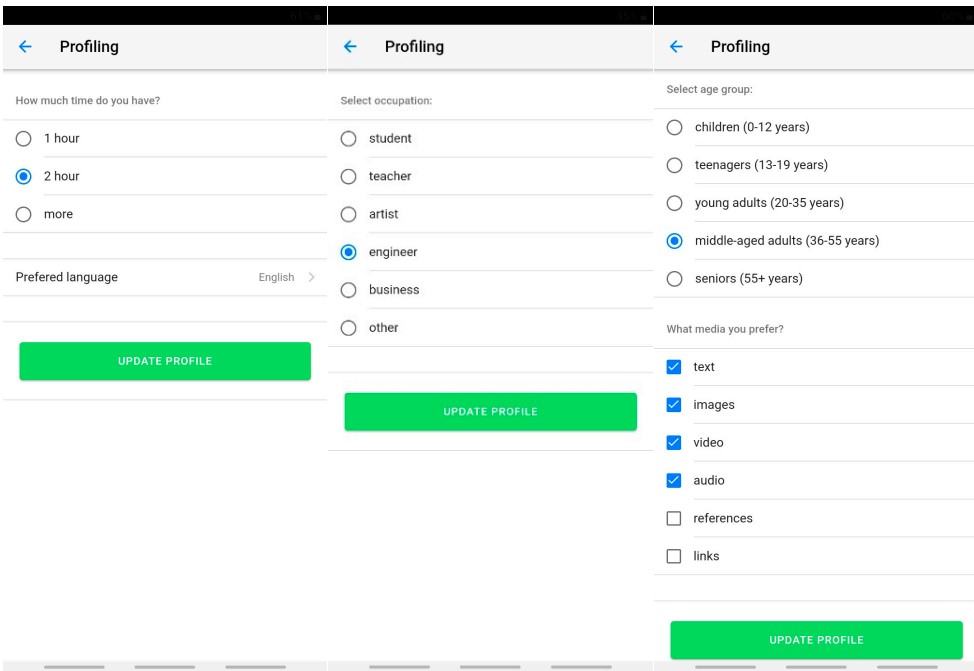

**Figure 10.** Explicit profiling.

### 4.4. Preliminary User Experience Test

User experience (UX) refers to the overall experience a person has when interacting with a product, system, or service, especially in terms of how easy, enjoyable, and effective the interaction is. It includes various aspects such as ease of use, aesthetics, accessibility, satisfaction, and emotional response.

Providing personalized content is critical to increasing visitor satisfaction and engagement at the museum. A mobile app that provides personalized exhibit information based on visitor preferences can significantly impact their overall experience. To ensure the success of such an app, thorough testing is essential to identify and fix any issues before the official launch.

The following section describes the preparation and testing of a beta version of a mobile application designed to provide personalized content to visitors of an open-air museum.

#### 4.4.1. Selection of Participants

To effectively evaluate the beta version of the service, a representative sample of museum visitors from different age groups needs to be selected. Based on statistics on the percentage distribution of museum visitors, the number of participants from each age group can be calculated as follows:

- Students: 21% of the total number of museum visitors.
- Adults: 50% of total museum visitors.
- Seniors: 29% of total museum visitors.

The number of museum visitors participating in the test was 24, including:

- Students: 5.
- Adults: 12.
- Seniors: 7.

#### 4.4.2. Quantifying User Experience

User experience can be quantified using a standardized UX questionnaire. In practice, the System Usability Scale (SUS) or User Experience Questionnaire (UEQ) are often used. These questionnaires use Likert scales to measure aspects such as perceived usability, efficiency, aesthetics, and satisfaction. The SUS scale is a quick and reliable usability

measurement tool. It consists of a 10-item questionnaire with five response options for respondents: from "strongly agree" to "strongly disagree". It can be used to evaluate software. The important advantages of this scale are that it gives reliable results in small samples and can effectively distinguish usable from unusable systems. The specific test uses questions specific to the service offered and questions from the SUS.

### 4.4.3. Quantifying User Experience

The survey aims to evaluate the user experience with the service to provide personalized content using the museum's mobile app. To provide concise and effective feedback, a 10-question survey has been developed. The questions will cover the following aspects:

- Relevance of personalized content: to assess whether the content delivered meets visitors' expectations and increases their engagement.
- Timing of content delivery: to determine whether the content is delivered at appropriate times during the visit.
- Improving visitor knowledge: assess whether the content contributes to improving visitor knowledge of the exhibits.
- Appropriate delivery through push notifications: to assess visitor satisfaction with the way information is delivered.
- User interface evaluation: to evaluate the usability and ease of use of the mobile application.
- Overall visitor satisfaction: to measure overall visitor satisfaction with the personalized content delivery service.

Based on the design of the user experience questionnaire, the following questions were defined:

1. Was the content delivered in line with your interests and expectations?—1 (strongly no) to 5 (strongly yes).
2. Were the notifications appropriately timed and not intrusive during your visit?—1 (not at all appropriate/intrusive) to 5 (extremely appropriate/not intrusive).
3. Rate the user interface of the mobile app and the interaction with it?—1 (extremely difficult to use) to 5 (extremely easy to use).
4. Has delivering content via push notifications improved your experience at the museum?—1 (not at all) to 5 (significantly improved).
5. Did the personalized content contribute to improving your knowledge of the exhibits?—1 (not at all) to 5 (significantly contributed).
6. How relevant and useful was the information provided by the museum's chatbot?—1 (not at all relevant/useful) to 5 (very relevant/useful).
7. How often did you use the museum's chatbot during your visit? (1—rarely or never, 5—very often).
8. Would you recommend the service for providing personalized content to others?—1 (strongly no) to 5 (strongly yes).
9. Would you answer the questions about creating your profile if it were optional—1 (strongly no) to 5 (strongly yes).
10. Overall, how satisfied are you with the service of providing personalized content through the museum's mobile app?—1 (very dissatisfied) to 5 (very satisfied).

### 4.5. Interpretation of Results

The number of participants in the test was 24. They were selected based on random sampling. The tests were conducted within one week. Each visitor who participated in the service testing completed an electronic survey after completing their visit within the museum. For each of the 10 questions on the survey, test participants were asked to give their rating from 1 to 5 on a Likert scale.

Figure 11 shows the results obtained after analyzing all surveys.

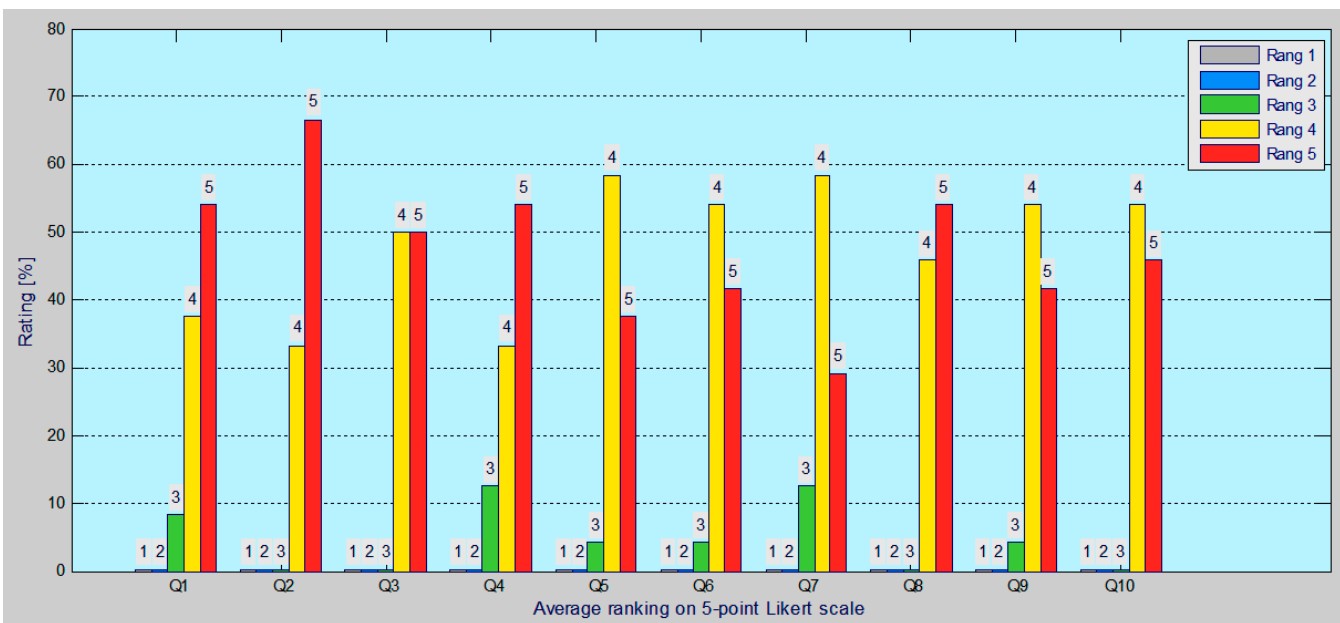

**Figure 11.** User's experience with the service.

Most participants (92%) in the test felt that the service provided content that met their interests and expectations (Q1); 54% of respondents answered, "strongly yes" and 38% answered "yes". Only 8% were undecided in their answer and answered neutral.

The service relies on delivering content about exhibits unobtrusively and only, when necessary, via push notifications. The results obtained for question 2 of the survey (Q2) are therefore very important—Were the notifications deployed at appropriate times and were they unobtrusive during your visit? As Figure 11 shows, most participants (67%) stated that the service provided content in an extremely unobtrusive manner and at the appropriate time (score 5), and 33% of them rated the service with a score of 4.

The user interface of the mobile app (Q3) scores were expectedly good. An equal number (50%) of respondents gave a rating of 5 and a rating of 4. The simplified user interface of the mobile app, as well as the lack of need for settings to use the app, suggested similar results.

Responses to question 4, whether delivering content via push notifications improved the visitor experience at the museum (Q4), garnered 88% high marks (4 or 5). Only 12% of respondents said they could not judge.

Personalization of content contributed to improving knowledge of exhibits (Q5) for 58% of respondents (score 4). Another 37% responded with a score of 5. A small proportion of respondents (5%) said they could not judge.

Only 5% of respondents said they could not judge how relevant and useful the information provided by the museum's chatbot was (Q6). Most respondents (54%) rated the chatbot a 4, with the remainder (41%) giving the highest rating of 5. These are very good ratings, and they suggest that the ChatGPT API can be successfully used for the purpose of delivering personalized content in museums. Similar are the responses to the question "How often did you use the museum's chatbot during your visit? (Q7). Out of all, 29% of the respondents used the chatbot very often (score 5), and 58% used it often (score 4); 13% gave a neutral response. There was no test participants who did not ask at least one question to the chatbot.

All respondents answered that they would recommend the service for providing personalized content to others (Q8).

Responses to question 9 (Q9) refuted our concerns that explicit profiling might fail if responses to survey questions were optional; 55% of respondents answered 'yes' to

this question and 42% answered 'strongly yes'. The reason for this is the way the survey questions are presented—in small portions, mostly after the visitor has viewed an exhibit.

Overall, respondents were satisfied (55%) or very satisfied (45%) with the service of providing personalized content through the museum's mobile app (Q10).

All these results are very encouraging for the feasibility and usability of the developed service to provide personalized content to museum visitors.

Comparison of Results from Different Age Groups

An analysis of the survey responses by age group was conducted. In the first age group "children and students" (6 to 20 years) the number of respondents was 5 (mean = 10.6, SD = 5.54). In the second age group "adults" (21 to 64 years old), 12 visitors participated (mean = 40.17, SD = 12.90). In the third age group "seniors" (over 64 years) 7 visitors participated (mean = 74.86, SD = 6.31).

Figure 12 shows the scores of the respondents grouped by age groups. The differences in estimates for question Q2 are also minimal. The highest score (4.80) was given by age group 1 participants and the lowest score (4.57) was given by age group 3 participants. This was to be expected as young people use push notifications from social networks daily.

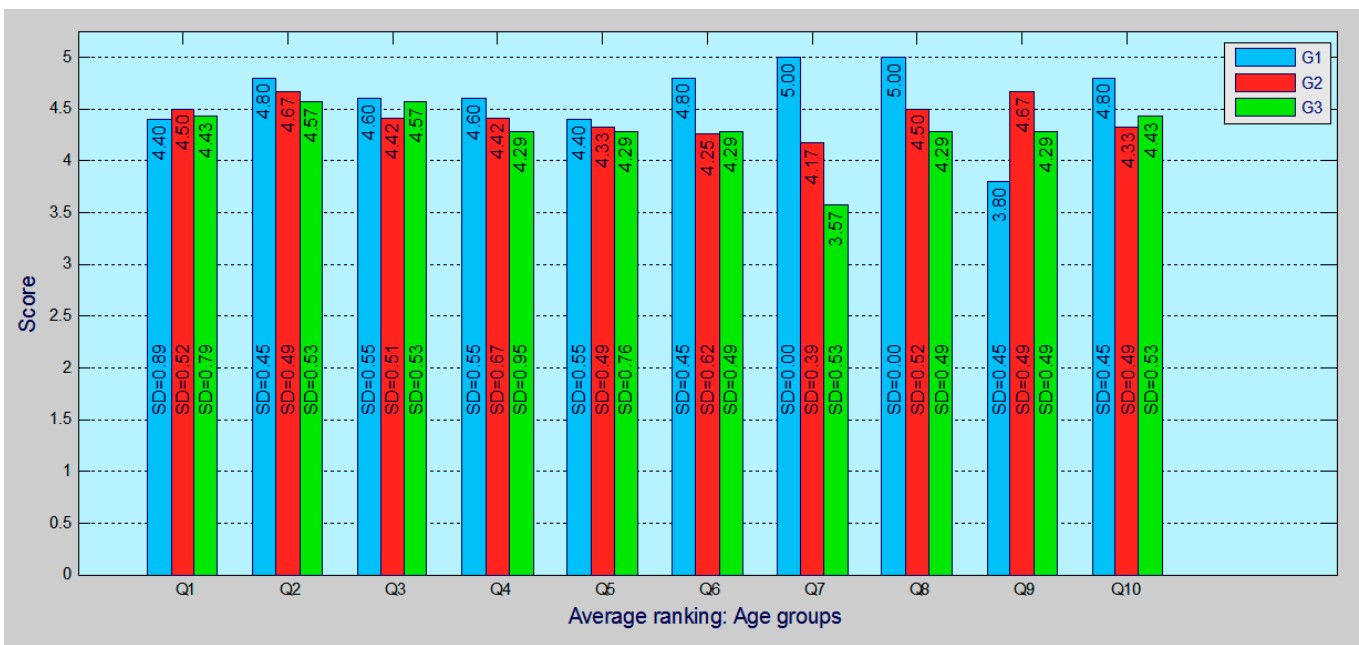

**Figure 12.** Users' experience in age groups.

When evaluating the user interface (Q3), the youngest and oldest gave slightly higher scores when compared to the scores of age group 2, 4.60 and 4.57, respectively. The intuitive interface, which displays the desired content without having to search for information through the mobile app menu, is equally convenient and usable for all age groups.

As expected, the youngest rated the effect of using push notifications to enhance their museum experience the highest (4.60). Visitors in age group 3 gave the lowest rating (4.29).

The scores for Q5 are very revealing. The difference in participants' ratings across the three age groups was statistically insignificant. This suggests that the choice of personalized content delivery method was equally well rated by all age groups.

The highest score (4.80) for the use of the museum's chatbot (Q6) was given by age group 1. In this age group, the number of questions asked through the chatbot for each exhibit ranged from 3 to 9. The scores of the other two age groups were also high, with 4.25 for group 2 and 4.27 for group 3. Only one visitor from age group 3 asked fewer than 5 questions about all the exhibits they viewed. This can be seen from the ratings of question

Q7 on how often visitors used the museum's chatbot. All visitors from age group 1 declared that they used the chatbot very often.

The most satisfied with the service (Q10) were visitors from age group 1. Their average score was 4.80. The scores of visitors from the other two age groups were relatively high at 4.33 and 4.43, respectively.

## 5. Discussion

The development of ExhibitXplorer presents several strengths that make it a valuable tool for personalized content delivery in museums. However, it also has some limitations that should be considered. This discussion describes the strengths and weaknesses of the system and discusses the potential implications it may have for museums.

### 5.1. Strengths

The evaluation results demonstrated the effectiveness and benefits of the ExhibitXplorer system in delivering personalized content. The following key findings were obtained:

- *Delivery of personalized content*: The system successfully delivered personalized content to different visitor segments based on their interests and preferences. Visitors reported high satisfaction and usefulness of the recommended content. Respondents from the researchers' segment appreciated the system's ability to provide in-depth information and references, and students found the system valuable for educational purposes. Casual visitors enjoyed the interactive and engaging content tailored to their interests. Personalized content is proactively delivered by analyzing visitors' proximity to geofences and their preferences.
- *Visitor segmentation*: The visitor segmentation algorithm showed good results. The implicit segmentation approach, which analyzed visitor interactions with the exhibits, accurately identified the interests and preferences of 16 out of 24 visitors. The remaining 8 visitors viewed the exhibits for too short a time and only explicit segmentation was used for them. Explicit segmentation, where visitors provided their preferences during a survey, complemented implicit segmentation, and further improved the accuracy of visitor profiling.
- *User satisfaction*: The overall user satisfaction rating of the ExhibitXplorer system is positive. Visitors appreciated the personalized content recommendations that enhanced their museum experience. The system's user-friendly interface, ease of navigation, and seamless integration with push notifications received high marks. Visitors also expressed satisfaction with the responsiveness of the system and the accuracy of exhibit information retrieved via the museum's chatbot.

The ExhibitXplorer system demonstrates good performance and scalability when working with real visitors. The distributed architecture combined with technologies such as Node.js and Python facilitated efficient processing of visitor requests and seamless communication between microservices. The system maintained high responsiveness and low latency, ensuring a smooth and uninterrupted user experience. Prior to testing with real visitors, the scalability of the system was validated through stress tests in which a significant number of concurrent visitor requests were simulated. The system demonstrated resilience and stability, handling the increased workload without compromising performance. The use of message brokers, such as RabbitMQ, facilitated asynchronous communication and contributed to the scalability and fault tolerance of the system.

### 5.2. Weaknesses

The weaknesses of the development can be sought in the following areas:

- *Visitor privacy and data security*: Since the system collects visitor data for segmentation and personalization purposes, ensuring visitor privacy and data security is of utmost importance. At this stage, visitors' personal data are not associated with their names and email addresses. Each visitor is identified by a unique string obtained upon successful registration to receive push notifications through the OneSignal service. In

addition, a UUID is retrieved for each mobile device on which the app is installed. This method of visitor identification guarantees the anonymity of their data, but there are drawbacks. If a user decides to uninstall the app, this will be detected by the OneSignal service. But if the same user decides to reinstall the application, he will receive a new identification code. The relationship between the old and new identifications is only the UUID of the mobile device. If the user activates the app on another mobile device, the link between the IDs is lost forever. Future enhancements to the service will link visitors' data to their email addresses, but will focus on implementing strong data protection measures, such as anonymization and encryption, to address privacy concerns and comply with data protection regulations.

- *Fine-grained visitor segmentation*: The current implementation of visitor segmentation focuses on broad visitor segments, such as researchers, students, casual visitors, etc. Future improvements could explore more fine-grained segmentation to provide even more tailored content recommendations. The use of visitor data on social networks can be a source of information that cannot be obtained through visitor profiling.
- *Integration with other services*: To enhance the overall museum experience, future improvements could include integrating the ExhibitXplorer system with other museum services, such as ticketing systems, guided tours, and interactive exhibits. This integration would provide visitors with a more immersive and comprehensive experience, allowing them to seamlessly navigate through different aspects of the museum visit.
- *Continuous user feedback*: Collecting and analyzing user feedback on an ongoing basis is crucial for understanding user needs, identifying areas for improvement, and enhancing the overall user experience. Regular surveys, feedback forms, and user interviews can provide valuable insights for refining the system and addressing user concerns.

## 6. Conclusions

The development of ExhibitXplorer represents a significant advancement in the field of smart museums, with far-reaching international implications. This innovative service addresses a universal need for personalized experiences in cultural and educational settings, transcending language and cultural barriers. By seamlessly integrating geofencing, artificial intelligence, and microservices, ExhibitXplorer has the potential to revolutionize how museums engage with diverse audiences worldwide. Its ability to cater to various visitor segments ensures that the benefits of tailored content delivery extend across a broad demographic spectrum. Furthermore, the inclusion of outdoor exhibits in the evaluation demonstrates ExhibitXplorer's adaptability to different museum environments, an aspect that is particularly relevant for open-air museums and heritage sites around the globe. The favorable user satisfaction ratings from the study emphasize ExhibitXplorer's contribution to the existing smart museum discourse, signifying its potential for broad adoption and influence on the global museum arena. The adoption of ExhibitXplorer can have significant effects on museums and their visitors. By tailoring the information to visitors' interests and preferences, the system promotes a more immersive and educational experience. The system's integration of AI, geofencing, and push notifications brings museums to the forefront of technological innovation, attracting a broader range of visitors, including younger generations. This technology-driven approach can rejuvenate the appeal of museums and contribute to their long-term sustainability.

ExhibitXplorer has demonstrated its effectiveness in delivering personalized content to museum visitors, enhancing their museum experience through tailored information and engagement. The system excels in delivering personalized content, owing to its distributed architecture and seamless integration of technologies. Nonetheless, there are opportunities for refinement and future development to bolster its capabilities and rectify its shortcomings. One notable avenue for future work is the integration of the GPT API 4.0, which would enable the analysis of exhibits directly from their photos. This advancement would open possibilities for customizing exhibit descriptions for the visually impaired and blind. By

leveraging image recognition and natural language processing, the system could generate audio descriptions or textual explanations of exhibits, making museums more inclusive and accessible to a wider audience.

However, it is important to acknowledge that the implementation of such a system requires careful consideration of ethical implications, privacy concerns, and visitor consent. Museums must ensure transparent communication with visitors regarding data collection and usage to maintain trust and respect visitor privacy.

In conclusion, ExhibitXplorer offers a promising solution for delivering personalized content in museums through its innovative approach, distributed architecture, and seamless integration of technologies. By leveraging the capabilities of the GPT API, the system enhances visitor engagement and satisfaction. Future work should focus on addressing privacy concerns, integrating accessibility features, and incorporating user feedback to further enhance the system's effectiveness and inclusivity.

**Funding:** This research was funded by the Bulgarian Ministry of Education and Science, Project № 2209E.

**Data Availability Statement:** The data are not publicly available due to research data ownership issues.

**Conflicts of Interest:** No conflict of interest exist in the submission of this manuscript.

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
