# Peer review of "ExhibitXplorer: Enabling Personalized Content Delivery in Museums Using Contextual Geofencing and Artificial Intelligence"

_ijgi, doi:10.3390/ijgi12100434_

Round 1

Reviewer 1 Report

The paper is generally interesting and its content and quality matches the journal.

The improvements that I would suggest are:

1. State early in the introduction the contributions of the paper. The intro can be shorter and some cited works can be moved to related work section.

2. Some more details are needed for the customisation of the content to the LLM. Is the model fine-tuned ? How is it done? Are there open source alternatives, and what are their properties? What would they require? Why was chatGPT selected over them?

Reviewer 2 Report

The paper under consideration provides an intriguing perspective on the use of the GPT API as a tool for generating personalized descriptions. The research addresses several critical considerations but presents some areas that require further attention and refinement.

Firstly, there is repetition of text in some sections. It is advisable to carefully revise the text to make the descriptions more streamlined and concise. This would significantly enhance the overall readability of the paper.

One key aspect the paper addresses is the utilization of the GPT API to obtain specific descriptions of art objects. The example provided regarding the Mona Lisa is valuable in illustrating the process. However, the paper would benefit from further expansion on how these descriptions are stored in the database and by whom they are generated. Does the user request specific information from the bot, and how is this information integrated into the process? This additional information would add a practical dimension and provide a more comprehensive understanding of the process.

Another crucial question raised is whether using the GPT API multiple times consistently yields the same result. This is a significant consideration, but the paper seems to touch on this point only superficially. It would have been beneficial to conduct a more in-depth experiment or assessment to establish the consistency of responses.

Finally, the paper briefly mentions that, in some cases, data from exhibits are used to train the Large Language Model (LLM) but does not provide sufficient details on whether it was created from scratch or subjected to a fine-tuning process. Additionally, it would have been helpful to delve further into the discussion regarding the quantity of data used for training, as the availability of high-quality data is crucial to the success of language models.

In conclusion, the paper lays a solid foundation and offers an interesting analysis of using the GPT API to improve the user experience in museums. I recommend acceptance with minor revisions to address the issues mentioned above.

Reviewer 3 Report

This paper is very interesting in the context of emerging artificial intelligence issues connected to museums. However, a revision is needed so that the paper to be publishable.

The first sentence of the paper mentions very well that museums play a vital role in preserving cultural heritage and educating the public. Here it can be mentioned several important sources in museum studies (see the book of Sodaro, A., 2018, the studies of Light D. on linking memorial museums to transitional justice, the study of Tamashiro, R and Furnari, E. on museums for peace, 2015).

Also, the introduction should present what this study brings new in existing museum studies.

The literature review is a bit too short and has to be engaged more with museum literature (for instance the role of museums as sites of trauma - see the study of Violi - doi - 10.1177/0263276411423035, the role of education for the younger people - see doi - 10.1080/14683857.2019.1702619, the effective turn in museums by Varutti M., 2023 - doi -10.1080/09647775.2022.2132993, how museums create empathy for their visitors - doi - 10.1080/15387216.2019.1581632, and so on).

The methods and results section are strong. However, limitation of data and method have also to be presented.

The discussion section presents the strenghts and weaknesses of the results, but it needs to be connected to the international literature on museum studies. Moreover, some policy recommendations could be highlighted at the end of the discussion or as a separate section..

Finally, conclusions should include one paragraph on the international implications of this study or how the outcomes of this study bring additional value to what we currently know in the (digital) museum literature.

Reviewer 4 Report

Dear editors and author,

Thank you for the opportunity to review this manuscript.

ExhibitXplorer is a new concept, and no previous article can be found online except the preprint from the author. Hence, this manuscript is innovative in a certain. However, I believe, this manuscript does not structure like a scientific paper but a product introduction or report. Neither the result, discussion, nor conclusion parts seem like an ordinary paper.

Besides, there are a few considerations the author may need to address. 

1. Missing citations.  For example, in the introduction part, the author demonstrates the importance of personalization in the museum experience (paragraph 3). Would you please add some reference for some ideas such as “Personalized content delivery enhances 45 visitor satisfaction…”, “Research has demonstrated that personalized museum visits lead to higher levels 47 of visitor satisfaction compared to generic experiences”, “When visitors feel that their interests and preferences are acknowledged and catered to, they are more likely to perceive their visit as valuable and enjoyable.”…and so on. Similar problems exist in many parts of this manuscript.

2. Wrong way of citing papers. For instance, in line 75, “In [10].

3. The introduction part of this manuscript is poorly constructed. There are 283 lines in this part, without any subtitles. Different contents jump randomly, and similar contents always appear more than once. For example, when the author talks about engagement, lines 55-65 already make it clear, and then line 84 discusses this again. When talking about satisfaction, the author already mentions it in line 48, but line 90 appears in this discussion again. This introduction fails to systematically present what the author wants to talk about in this manuscript.

4. Many “We”, and “the authors” appear in this manuscript, although this is a solo author manuscript.

5. It would be better to present “The algorithm for generating personalized content is as follows:” by figure.

Hope these comments will help the author improve this manuscript.

Best wishes,

Reviewer

Minor editing of English language required

Round 2

Reviewer 3 Report

Authors have much improved their paper, so I am happy to propose this paper to be accepted for publication.

Reviewer 4 Report

Dear author and editor,

I believe this manuscript has significantly improved. However, the introduction still can not meet my expectations even after the author deleted quite a large part of it. This part fails to demonstrate the importance and research gaps of personalized content as well as artificial intelligence, while only the basic knowledge of the museum is left.
